# A Comparison of Vaccine Hesitancy of COVID-19 Vaccination in China and the United States

**DOI:** 10.3390/vaccines9060649

**Published:** 2021-06-14

**Authors:** Taoran Liu, Zonglin He, Jian Huang, Ni Yan, Qian Chen, Fengqiu Huang, Yuejia Zhang, Omolola M. Akinwunmi, Babatunde O. Akinwunmi, Casper J. P. Zhang, Yibo Wu, Wai-Kit Ming

**Affiliations:** 1Department of Public Health and Preventive Medicine, School of Medicine, Jinan University, Guangzhou 510632, China; t.liu.10@student.rug.nl (T.L.); yanni666@stu2019.jnu.edu.cn (N.Y.); chenqian@stu2020.jnu.edu.cn (Q.C.); fengqiu@stu2020.jnu.edu.cn (F.H.); 2International School, Jinan University, Guangzhou 510632, China; hezonglin0leon@stu2015.jnu.edu.cn; 3Singapore Institute for Clinical Sciences (SICS), Agency for Science, Technology and Research (A*STAR), Singapore 138632, Singapore; jian.huang@imperial.ac.uk; 4Department of Epidemiology and Biostatistics, School of Public Health, Faculty of Medicine, Imperial College London, London W2 1PG, UK; 5School of Medicine, Jinan University, Guangzhou 510632, China; zhangyuejia69@stu2019.jnu.edu.cn; 6Department of Radiology, College of Medicine, University of Ibadan Nigeria, Ibadan, Nigeria; omololakinwunmi@gmail.com; 7Department of Radiology, University College Hospital, Ibadan, Nigeria; 8Department of Obstetrics and Gynecology Brigham and Women’s Hospital Boston, Boston, MA 02115, USA; bakinwunmi@bwh.harvard.edu; 9Center for Genomic Medicine (CGM), Massachusetts General Hospital, Harvard Medical School Harvard University, Boston, MA 02115, USA; 10School of Public Health, The University of Hong Kong, Pokfulam, Hong Kong, China; casperz1@connect.hku.hk; 11School of Public Health, Peking University, Beijing 100191, China; 12Key Research Base of Philosophy and Social Sciences in Shaanxi Province, Health Culture Research Center of Shaanxi, Xi’an 712046, China

**Keywords:** global health, COVID-19, vaccine hesitancy, public health, health policy, vaccine preference

## Abstract

Objectives: To investigate the differences in vaccine hesitancy and preference of the currently available COVID-19 vaccines between two countries, namely, China and the United States (U.S.). Method: A cross-national survey was conducted in both China and the United States, and discrete choice experiments, as well as Likert scales, were utilized to assess vaccine preference and the underlying factors contributing to vaccination acceptance. Propensity score matching (PSM) was performed to enable a direct comparison between the two countries. Results: A total of 9077 (5375 and 3702 from China and the United States, respectively) respondents completed the survey. After propensity score matching, over 82.0% of respondents from China positively accepted the COVID-19 vaccination, while 72.2% of respondents from the United States positively accepted it. Specifically, only 31.9% of Chinese respondents were recommended by a doctor to have COVID-19 vaccination, while more than half of the U.S. respondents were recommended by a doctor (50.2%), local health board (59.4%), or friends and families (64.8%). The discrete choice experiments revealed that respondents from the United States attached the greatest importance to the efficacy of COVID-19 vaccines (44.41%), followed by the cost of vaccination (29.57%), whereas those from China held a different viewpoint, that the cost of vaccination covered the largest proportion in their trade-off (30.66%), and efficacy ranked as the second most important attribute (26.34%). Additionally, respondents from China tended to be much more concerned about the adverse effect of vaccination (19.68% vs. 6.12%) and have a lower perceived severity of being infected with COVID-19. Conclusion: Although the overall acceptance and hesitancy of COVID-19 vaccination in both countries are high, underpinned distinctions between these countries were observed. Owing to the differences in COVID-19 incidence rates, cultural backgrounds, and the availability of specific COVID-19 vaccines in the two countries, vaccine rollout strategies should be nation-dependent.

## 1. Introduction

A pneumonia-like disease outbreak, now named the 2019 coronavirus disease (COVID-19), caused by a newly identified coronavirus, severe acute respiratory syndrome (SARS)-CoV-2, swept the globe in early 2020 [1]. Although its exact origin remains unknown, and detailed knowledge of its transmission is still limited, this global pandemic has become the most serious public health threat from a respiratory virus since the 1918 H1N1 influenza pandemic [2,3]. One year after the onset of the pandemic [4], COVID-19 is continuing to impose tremendous burdens on the public health systems and economies globally [5,6]. As of early March 2021, 219 countries or regions have reported confirmed cases [7], with over 120 million confirmed cases and over 2.66 million deaths, and a case fatality rate of over 2.21% worldwide [8].

Various public health measures, specifically, non-pharmaceutical interventions (NPIs) that showed effectiveness in previous infection outbreaks (e.g., mass facemask use, social distancing, and home quarantine), have been implemented by governments to contain the spread of SARS-CoV-2 [2,9,10]. Although the effectiveness of these public health measures in this outbreak remains to be determined, these NPIs may carry a high economic cost, leading to a long war of attrition in society [11].

Therefore, massive vaccination coverage is considered as a prerequisite to achieve herd immunity and therefore curb the COVID-19 pandemic [12]. By March 2021, several COVID-19 vaccines [13,14] had been developed, and vaccination rollouts had started in a few countries. However, people remain uncertain of the safety and efficacy of the vaccines. This becomes extensive, especially following reports of adverse events after shots of the mRNA vaccination [15], sudden death secondary to shots of mRNA COVID-19 vaccines in Norway [16], and incidences of inactivated vaccines during a phase 3 trial in Brazil, among others [17,18,19]. Additionally, according to a national poll in the United States, only 58% of adults aged from 50 to 80 are willing to receive the COVID-19 vaccine [20]. Additionally, as increasing number of variants have been identified around the globe [21,22], the current vaccines may no longer provide effective protection against the SARS-CoV-2 virus, including the variants.

Some studies have found that COVID-19 vaccine acceptance varied to a large extent across countries and regions before the vaccine became available; the vaccine hesitancy of COVID-19 is increasing globally [23,24]. This may influence the vaccination coverage and hinder the establishment of herd immunity. Interestingly, increasing vaccine hesitancy is being noticed among Chinese residents and some Asian countries where the transmission of COVID-19 has been well-controlled, specifically, when the vaccines are completely free in some countries, and such phenomena may be attributable to the low perceived benefits of the vaccines and low perceived risks of the COVID-19 pandemic, and most importantly, the low perceived efficacy and the increasingly high skepticism of the efficacy and safety of the currently available COVID-19 vaccines [25]. Therefore, it is urgent to find the hindering factors and further promote the COVID-19 vaccination to achieve herd immunity.

Vaccine hesitancy is multifactorial, attributable to general misinformation spread, perceived risks of disease, perceived efficacy and safety of vaccines, attitudes and demand of vaccines, cultural and religious factors and other unspecific factors [26,27]. China and the United States (U.S.) are two of the representative countries that are among the countries hardest hit by the pandemic [28], and both have a large population but very different cultural and religious systems [29]; hence, populations’ acceptance to COVID-19 vaccination of the two countries could be strategically significant for further vaccine promotion and herd immunity.

Nevertheless, direct comparison between countries could be biased and irrelevant owing to the high heterogeneity and confounding effects; hence, in the present study, propensity score matching (PSM) was used to compare the two populations. The following aims were addressed: (1) to investigate the census-level acceptance and preference of residents and further explore the influencing factors underlying their decision-making towards COVID-19 vaccination; and (2) to compare the two countries in a statistically comparable way to demonstrate the vaccine hesitancy in both countries and further provide insights into future strategies of large-scale COVID-19 vaccination coverage in the two countries.

## 2. Materials and Methods

### 2.1. Study Design

An anonymous, self-administered, cross-sectional survey was conducted online in China and the United States through multiple international online panel providers (for data collecting in the United States) and recruited volunteers across China (for data collecting in China). Stratified sampling by age and geological locations was used, and nationally representative samples of the general adult populations were collected in China, while the U.S. respondents were mainly collected via MTurk [30]. The questionnaire was established in Lighthouse studio (Sawtooth Software, version 9.8.1). The study was approved by the Jinan University Institutional Review Board. In the survey, participants responded to a total of 55 items, with standard demographic and socio-economic questions including age, sex, level of education, annual income, and marital status, followed by one set of discrete choice experiments (DCEs) and questions about risk perception, COVID-19 impact, attitudes, and acceptance of and attitudes towards vaccines against COVID-19 during the pandemic. Prior to formal data collection, a pilot study in China had been conducted to evaluate the content validity and the reliability of the questions with both experts and general populations, and a group of experts was consulted to improve semantics and readability.

### 2.2. Respondents

The inclusion criteria were respondents aged 18 years and above without cognitive impairments (self-report). Respondents were randomly recruited and selected through multiple international online panel providers (MTurk and Dynata) and by nearly 100 experienced volunteers recruited across China using a stratified sampling method. No personally identifiable information was collected. A total of 12,959 respondents were recruited, with a total of 9077 respondents (5375 and 3702 respondents from China and the United States, respectively) included in our study (Figure 1).

### 2.3. Data Collection

The survey data were collected between 29 January and 13 February 2021. The respondents were required to provide a randomly generated code after completing the survey to ensure that they were real people and not robots. All the questions were close-ended, with tick boxes provided for responses and no question skipping allowed, and no data would be stored if the website of the questionnaire was closed before completion. Hence, no missing data were generated.

General acceptance was defined as scoring higher than 6 with regard to answering the question “How do you rate your willingness and acceptance to get COVID-19 vaccination? (if the vaccines are generally available)”, and the *acceptance under social cues* was defined as scoring higher than 6 for the acceptance of COVID-19 vaccines if the vaccination was recommended by the respondents’ family members, friends or employers. Countries were categorized as developed and developing countries according to the United Nations country classification [31]. Educational level was further classified into four groups, where “low” signified respondents reporting having not finished secondary education (high school); “medium” signified those who had completed secondary, vocational, or equivalent degrees; the “high” group consisted of those who had completed a tertiary or bachelor’s degree; and “very high” indicated postgraduate work. Moreover, two external COVID-19-related variables were added, as per the method reported by Lazarus et al. [22], i.e., the total COVID-19 positive cases per million persons and total COVID-19 deaths per million persons as reported by Worldometer on 21 January 2021 [32]. For COVID-19 cases per million in the population, “High” was defined as more than 10,000 cases per million people, “medium” as between 1000 and 10,000 cases per million people, and “low” was defined as below 1000 cases per million people. For COVID-19-specific mortality cases per million in the population, “High” was defined as more than 1000 deaths per million people, “medium” as between 100 and 1000 deaths per million people, and “low” was defined as below 100 deaths per million people.

### 2.4. Survey Design

The questionnaire consisted of three sections with a total of 55 items that required a response. In the first section, respondents responded to provide socio-demographic information regarding age, sex, educational level, occupation, income level, nationality and marital status. Additionally, respondents were required to rate their willingness and acceptance to be vaccinated from “totally unwilling” to “totally willing” with and without social cues. These questions were, “How do you rate your willingness and acceptance of getting vaccinated?” and “How do you rate your willingness and acceptance if your friends, family members, and neighbors recommend you do so?”, respectively. Additionally, respondents responded to questions related to COVID-19 infection history as well as the major sources of information with regard to COVID-19 vaccines.

In the second section, a DCE was used to further explore respondents’ preference for the currently available vaccines. Specifically, vaccine attributes and their levels were identified and retrieved through the relevant literature and several vaccines on the market, and the attributes were then ranked, categorized, and refined by a group of experts in the field of public health and vaccination [14,15,16,17,18,33,34,35,36]. A total of six attributes were identified, and a two-vaccine profile was randomly adopted for the DCE design. Detailed descriptions of the attributes and levels are given in Appendix A. During the survey, respondents were asked to make a series of hypothetical choices and estimate their preference for different attributes of the vaccine, based on various scenarios.

In addition, to assess respondents’ attitudes toward COVID-19 and the acceptance of COVID-19 vaccination and their influencing factors, questions were designed based on the five-concept health belief model and other frameworks, as reported by various previous studies to assess vaccine acceptance and hesitancy for newly emerging infectious diseases such as H1N1, MERS, or Ebola [37]. The following content was included in this section: (1) the perceived susceptibility to COVID-19 infection; (2) perceived severity of COVID-19 infection; (3) perceived benefits of COVID-19 vaccination; (4) perceived barriers to COVID-19 vaccination; (5) cues to action for COVID-19 vaccination, such as a recommendation from a doctor or local health board, were assessed; (6) socio-economic factors; and (7) past immunization behaviors. Most questions were assessed on a seven-point Likert scale. The reliability (alpha = 0.8951) and validity (Kaiser–Meyer–Olkin Measure = 0.942) of the Likert scale were tested.

### 2.5. Statistical Analysis

The data in this study were collected in an international cross-sectional survey. To minimize potential confounding biases due to discrepancy in baseline characteristics, propensity scoring was calculated and matched to balance covariates for respondents in China and in the United States. Propensity score matching is a statistical technique that can help strengthen causal arguments in quasi-experimental and observational studies by reducing selection bias [38]. The covariates were identified using the pair-wise Pearson correlation matrix, and the results are presented in Appendix A. Subsequently, the set of covariates was determined by minimizing the residual confounding factors as much as possible, where a logistic regression model was performed to estimate the propensity scores for each group of respondents. Finally, a total of 3436 respondents, with half from China and the other half from the United States, were matched using propensity scores from the total 12,959 respondents, with the covariates being sex, age, and annual income.

Descriptive statistics were analyzed to describe the characteristics of socio-economic status and demographic information, risk perception, pandemic impact, as well as acceptance, attitudes and preferences of COVID-19 vaccines, using central tendency (mean, median) and dispersion (standard deviation, interquartile interval) measures. The chi-squared test or Fischer’s exact test was used for the univariate analysis of qualitative variables, and Student’s *t*-test or Mann–Whitney test for quantitative variables. Absolute and relative frequencies are presented for qualitative variables, while quantitative variables are presented as means (standard deviation). Multivariate logistic regression was then performed between the vaccine demand group and vaccine delay group to identify the influencing factors of vaccination acceptance (immediate or delayed acceptance), with the odds ratio (OR), standard error (SE), and a 95% confidence interval (CI) being calculated. The data were analyzed using STATA, version 14.0 (Stata Corp, College Station, TX, USA). For the DCE part, we employed a conditional logit model (CLOGIT) to quantify respondents’ preference for vaccines’ attributes and levels in their trade-off in general, and to further explore participants’ preference heterogeneity among different countries and regions. After the conditional logit model, we dummy-coded all the attribute levels, and after dummy coding, the model parameter *β* represented the value that respondents placed on an attribute level relative to the reference level, where the model parameter *β* does not directly reflect the preference weight within an attribute; this presentation can enhance the interpretation of the preference weights specifying the difference between two random coefficients. The data of the DCE part were analyzed using STATA, version 14.0 (Stata Corp, College Station, TX, USA).

### 2.6. Scenario Analysis and Simulation

We also performed a scenario analysis and product simulation in Lighthouse studio (version 9.9.1) to further explore vaccines with which characteristics contributed most to respondents’ preferences and with the highest probability for acceptance. The base scenario was hypothesized based on the vaccine variety, with all the attributes’ levels being the lowest utilities (except cost attribute), whereas the best scenario was assumed to have each attributes’ level at the highest utilities (except cost attributes). Additionally, the other scenarios were established according to the currently available information of vaccines from various clinical trials. We use the share of preference as our simulation model, because this model helped us to better predict the level of preference any vaccines might achieve. The whole simulation was performed in two steps: (1) subject the respondent’s total utilities for the product to the exponential transformation, specifically as s=exp(utility); and (2) rescale the results to a total of 100% [39].

## 3. Results

### 3.1. Respondents’ Characteristics

The present study conducted a large-scale self-administered online survey in China and the United States. A total of 9077 respondents (5375 and 3702 respondents from China and the United States, respectively) completed the survey and were selected and further analyzed in the present study. Concerning the pre-PSM samples, the respondents from the United States tended to be more highly educated and earning more money annually compared to respondents from China. For the two respondent groups, the majority of the respondents were female (55.3% for China, 51.8% for the United States), and 47.8% of the respondents in China and 67.6% of those in the United States held a bachelors’ degree or higher. After PSM for age, sex, education, annual income, and occupation, no statistically significant discrepancies could be discerned between the respondents from the two countries in demographic characteristics (*P* = 1.00 for age, sex, education, annual income and occupation).

### 3.2. Generate Hesitancy and Participants’ Vaccination History

As the pre-PSM results listed in Table 1 and Figure 2 show, respondents from China had a relatively lower hesitancy of COVID-19 vaccines (7.8/10) than those from the United States (7.2/10), when asked, “How do you rate your willingness and acceptance to get a COVID-19 vaccination? (if the vaccines are generally available)” (*general acceptance*). For post-PSM, over 82.0% of respondents from China positively accepted the COVID-19 vaccination, while only 72.2% of respondents from the United States positively accepted it, and the proportions changed to 83.3% and 71.0%, respectively, if the respondents were recommended the vaccination by friends, family members, employers, etc. Additionally, around 12% of respondents from China and 38.3% from the United States had delayed or canceled vaccinations for reasons other than illness or allergy.

Interestingly, post-PSM, 31.9% of Chinese respondents were recommended by a doctor to receive a COVID-19 vaccination, whereas more than one-half of the U.S. respondents (50.2%) were recommended by a doctor (Appendix A). Additionally, generally, more than half of the U.S. respondents were recommended to receive a vaccination, either by the local health board (59.4%) or by friends or families (64.8%).

The influence of demographic factors on the average rating of the hesitancy of COVID-19 vaccination between the two countries is shown in Appendix A, and detailed subgroup analyses of the hesitancy by sex, age intervals, education, occupation, and annual income are shown in Appendix A, respectively. Additionally, the proportion of vaccine hesitancy from China was generally smaller than those from the United States, except for the respondents with a Master’s academic degree, or with an annual income over USD 70,000, or with skilled, agricultural, forestry and fishery-related occupations. In both China and the United States, respondents’ vaccination hesitancy increased with the academic degree above bachelor’s degree. Moreover, being female and more highly educated, as well as themselves being infected or immediate friends or families being infected, served as a contributor for the respondents hesitating before vaccination, as shown in Appendix A.

### 3.3. Post-PSM Participants’ Vaccine Preference, Attributes and Level of Importance

The relative attributes’ importance compared between the United States and China is shown in Figure 3. After PSM, we found that respondents from the United States attached the greatest importance to the efficacy of COVID-19 vaccines (44.41%), followed by the cost of vaccination (29.57%), whereas those from China held a different viewpoint: the cost of vaccination covered the largest proportion in their trade-off (30.66%), and efficacy ranked as the second most important attribute (26.34%). Additionally, respondents from China were also much more concerned about adverse effects of the vaccines, which was ranked as the third most important factor (19.68%). The duration of vaccination, vaccine varieties and time for the vaccine starting to work remained of relatively low importance in both countries.

Interestingly, in the post-PSM results, respondents from the United States preferred the mRNA COVID-19 vaccines, whereas respondents from China preferred the inactivated COVID-19 vaccine (OR = 1.164, 95% CI (1.124, 1.205), *p* < 0.001). Additionally, respondents’ vaccine preference significantly decreased with a moderate adverse effect compared with a very mild adverse effect. In both countries, respondents’ preference increased along with the rise of efficacy and reached a peak at 95% efficacy (vs. an average of 55%). The reduction in vaccine preference only appeared in respondents from the United States if the time for the vaccines to start working increased (20 days vs. 5 days) (OR = 0.893, 95% CI (0.849, 0.938), *p* = 0.06). Moreover, respondents from both countries preferred a longer vaccine protection time and a low level of vaccination cost.

The post-PSM preference comparison between sex in both countries is shown in Appendix A. For respondents from the United States, both male and female respondents ranked the efficacy of COVID-19 vaccines as an essential attribute; the efficacy was slightly more important for females (male: 43.12%, female 45.07%). For respondents from China, the cost of vaccination had the greatest relative importance and was slightly more important for females (male: 28.85%, female: 31.45%). Similarly, the efficacy of COVID-19 vaccines also was valued more by female respondents than male respondents in China (male: 43.12%, female: 45.07%) but more by male respondents than female in the US (male: 26.36%, female: 26.02%).

### 3.4. Post-PSM Scenario Analysis and Uptake Likelihood Prediction

Table 2 presents the simulated share of preferences under nine different scenarios, which were based on clinical trials and real-world data of various COVID-19 vaccines, as reported by various large-scale clinical trials. The base scenario is the vaccine with seemingly lower preference, and only 2.9% of the respondents from the United States were willing to accept the base scenario vaccine, whereas 4.5% of those from China preferred to take the base scenario vaccine. However, we noticed that the U.S. respondents were more likely to choose mRNA vaccines (share of preference 14.2% for Scenario 2 and 17.3% for Scenario 3), whereas those from China placed a stronger preference on an inactivated vaccine (Scenario 4, share of preference 13.7%). However, the respondents from the United States also had a higher preference for the adenovirus vaccine (Scenario 5), and the Chinese respondents had a higher preference for mRNA vaccine too (Scenario 3, share of preference 13.4%). If all the attributes of the vaccine were set as the best levels, as indicated in the MNL analysis (Scenario 8), then 19.1% and 17.6% of the respondents from the United States and China, respectively, would choose this hypothetical vaccine.

### 3.5. Behavioral and Psychological Results

The Likert scale, as presented in Appendix A, indicated that, generally, the respondents had a highly positive attitude towards the benefits of COVID-19 vaccines (United States: 15.8/21; China: 17/21) and were not so concerned about the risks and barriers of COVID-19 vaccination, although respondents from China were slightly more concerned about risks than those from the United States (United States: 11.3/21; China: 13.3/21). Additionally, the respondents generally believed in the necessity and efficacy of vaccination in the prevention of diseases, because they highly rated the item, “In general, vaccination is effective in preventing diseases” (United States: 5.5/7.0; China: 5.7/7.0), followed by scores of 5.9/7.0 and 5.3/7.0 for respondents from China and the United States, respectively, for the item, “In general, prevention is better than cure”. In terms of socio-cultural factors, religion- and gender-related reasons were the least contributory to the decision-making regarding COVID-19 vaccination (United States: 3.1/7.0; China:4.2/7.0 for religious or cultural reasons, and United States: 4.4/7.0; China: 5.7/7.0 for gender reasons). Respondents rated the following item the lowest score: “I believe that people are risking their health or the health of the society if they do not take a COVID-19 vaccine” (United States: 2.5/7.0; China: 3.5/7.0).

## 4. Discussion

The present study sought to provide a comprehensive investigation of the acceptance of COVID-19 vaccination in China and the United States and compare the hesitancy and preference of COVID-19 vaccines between the two countries, controlling for demographic characteristics using PSM. The unmatched samples of the respondents from the two countries were adjusted with the uncontrolled quota sampling method. Hence, the results may demonstrate a census-level investigation in two countries. Generally, the majority of the respondents in both countries had high acceptance regarding COVID-19 vaccination, either general acceptance or acceptance under recommendation from friends, families or employers.

In our study, we found that respondents from China had a relatively higher acceptance than those from the United States, and when compared to studies conducted before the vaccine became available, or even before the results of clinical trials on some COVID-19 vaccines were published, their acceptance also increased [40]. Existing opinion polls for the acceptance of hypothetical COVID-19 vaccines by U.S. citizens ranged from 40% to 70% [20,41,42,43]. Additionally, in a population-based survey conducted in Hong Kong, the overall vaccine acceptance rate was only 37.2% [44], whereas more than 90% of respondents in mainland China would like to be vaccinated for COVID-19 vaccines, regardless of the efficacy [40]. Although it has been reported that some perceived susceptibility to infection or perceived risks may not be associated with COVID-19 vaccine acceptance [44], our results show that there is a distinct difference between the respondents from China and the United States regarding the perceived risks of infection and perceived benefits of vaccination after propensity score matching. The Chinese respondents tended to be more concerned about the health problems COVID-19 may cause to them if they were infected, whereas the U.S. respondents were more concerned about being infected (Appendix A). Additionally, some studies have found an association between the incidence of a diseased population and perceived risks, and low-risk perception was also associated with the respondents trusting health professionals and health officials for information on COVID-19 [45].

Public concern about vaccine safety and efficacy has frequently been reported as one of the major obstacles to vaccination acceptance, specifically when the COVID-19 vaccines were being developed and rolled out at an unprecedented speed [46,47]. However, it is notable that, despite high acceptance of the vaccines, Chinese respondents were more concerned about the safety of the vaccines; the adverse effect of the COVID-19 vaccines had far more relative importance for respondents from China than those from the United States. Combined with their low perceived risks of susceptibility, this may therefore lead to their placing greater importance on the attribute “cost” in the discrete choice experiment (Table 3 and Figure 2), and may also partially explain why the vaccination rate remains at a relatively low level in China, compared with some other countries in the world, such as the United States, the United Kingdom, etc.

Although both populations from the United States and China highlighted the effect of cost in their preferences, both governments have made relevant policies to guarantee that COVID-19 vaccines are free of charge. Therefore, the effect of the cost of COVID-19 vaccines can be ignored to some extent when promoting vaccination. As of 31 March 2021, a total of 114.69 million doses of COVID-19 vaccines have been reported in China; however, the vaccination rate remains below 10% [48]. There is still a huge gap before achieving herd immunity, which some experts have estimated would require a 70% to 90% vaccination rate [49,50]. In contrast, the vaccination rate in the United States has reached 30% (of those who have received at least one dose) [51], with 103.35 million people receiving one or more doses cumulatively. Therefore, although the vaccination growth rate seems to have been more rapid in China than in the United States from 24 March 2021, the speed is not rapid enough to achieve herd immunity in the short term; eventually, the relatively low level of vaccination in China may force China to lose its advantage in COVID-19 epidemiological control [52]. Therefore, improving the speed of vaccination is currently a top priority for all countries around the world to control and ultimately eradicate the pandemic.

In both pre-PSM and post-PSM samples, we found that the respondents from China attached more importance to cost, whereas those from the United States attached more to efficacy. This finding may strongly correlate with local COVID-19 incidence and mortality rates, i.e., the perceived risk for the public to become infected. As of 14 March 2021, there have been over 120 million cumulative COVID-19 cases and around 600 thousand cumulative deaths [8] in the United States. Thus, vaccines with high efficacy are urgently needed to contain the development of epidemic in the United States, whereas in China, there have been over 102,000 cumulative cases and around 5000 deaths [8]. Additionally, we found that respondents from the United States preferred to be vaccinated with mRNA vaccines, whereas respondents from China considered inactivated vaccines as the best choice; this might be related to the actual availability of the types of vaccines in the United States and China. High vaccination coverage depends on public understanding of the need and value of vaccination, the availability of vaccines, as well as accessible immunization services [53]. Additionally, to further promote high vaccination coverage, the Strategic Advisory Group of Experts (SAGE) on Immunization Working Group on Vaccine Hesitancy recommends three categories of strategies, namely, an increase in the understanding of vaccine hesitancy, the enhancement of structural and organizational capacity at global, national and local levels, and international collaboration between countries regarding the development, validation, and implementation of new tools to address hesitancy [53,54]. Additionally, our study emphasizes the key role of international collaboration. Formal collaboration has been institutionalized through the World Health Organization (WHO); however, in the contemporary COVID-19 pandemic, many countries have demonstrated poor performance in tackling COVID-19, with reflections of strong, self-interested nationalism [55]. Vaccine nationalism has greatly restricted international collaboration and further hindered epidemic control. Meanwhile, vaccine nationalism has been warned by WHO to be a moral failure. Therefore, to relieve the global epidemic, collaboration on COVID-19 vaccines has been urgently needed, especially when some adenovirus COVID-19 vaccination programs have been suspended due to adverse effects [56].

Our findings are useful for designing and modifying effective vaccination promotion strategies and immunization coverage programs for the public and those with vaccine hesitancy, based on different national conditions and contextual backgrounds. First, the perceived risks of infections and risks of severe health problems secondary to the infection may be vital for the respondents to actively seek vaccination. Hence, the risks of infection and adverse outcomes secondary to the infection should be clearly outlined by the media or governments to the public to enhance mutual understanding. Secondly, the vaccine prices should be affordable and available for the public; it is beneficial that many countries have financed their immunization programs, making COVID-19 vaccines free of charge for the public [57,58,59]. Moreover, monitoring information about vaccine safety should be published on a regular basis after the application of the vaccine, and timely health education and communication should be conveyed by authoritative sources [43,60].

China and the United States have plenty of differences, especially in contextual influences, for instance, in the media environment, influential leaders, historical influences, policies, and cultural factors. The majority (82.0% of Chinese respondents, 72.2% of U.S. respondents) of respondents generally accepted the COVID-19 vaccination; therefore, it is still worth further exploring and identifying other barriers or facilitators to their vaccination decisions based on different contextual backgrounds and different countries.

Although the average COVID-19 vaccine acceptance of respondents from the United States may be lower than those from China, higher annual income was associated with higher vaccine hesitancy among United States respondents. Respondents with higher educational levels may also tend to acknowledge more vaccine risks.

### Strengths and Limitations

The present study has some limitations. The inherent nature of cross-sectional studies renders it difficult to establish causality or to generalize the results in a long-term manner, especially when vaccination acceptance is variable, dynamic and multifactorial. Hence, the results of the study should be interpreted with caution. However, the present study implemented the PSM to minimize the confounding effects, and it may somehow enhance the interpretability of the results. Moreover, in view of impartiality, we did not collect information regarding race and ethnicity in the survey. Therefore, racial classifications of the respondents were not reported or considered in the analysis in the present study. Regarding survey distribution, the present study utilized multiple means to collect data, where the Chinese respondents were selected using stratification sampling via age and geographical location, while the U.S. respondents were sampled mainly through MTurk. Even though we have evaluated the reliability of the data and sampling adequacy using Kaiser–Meyer–Olkin analysis (KMO coef = 0.8958), and that several studies have reported the reliability of census-level data retrieved from MTurk, the results should still be interpreted with caution.

However, the present study is the first to directly compare the acceptance and preference of respondents from two distinctly different representative countries, and the results of this study may provide insights for vaccination promotion strategies based on different national situations globally. Additionally, the present study used multiple study methods to provide the most comprehensive and updated investigation, especially on the influencing factors contributing to the decision-making regarding vaccination.

## 5. Conclusions

In conclusion, great variability in the preference of COVID-19 vaccines was found between respondents from China and the United States, and the influencing factors for hesitancy varied, as did the attributes for their preferences for the vaccines; hence, multi-disciplinary and international collaboration should be established and strengthened based on specific national conditions to further reduce vaccine hesitancy and increase vaccination coverage.

## Figures and Tables

**Figure 1 vaccines-09-00649-f001:**
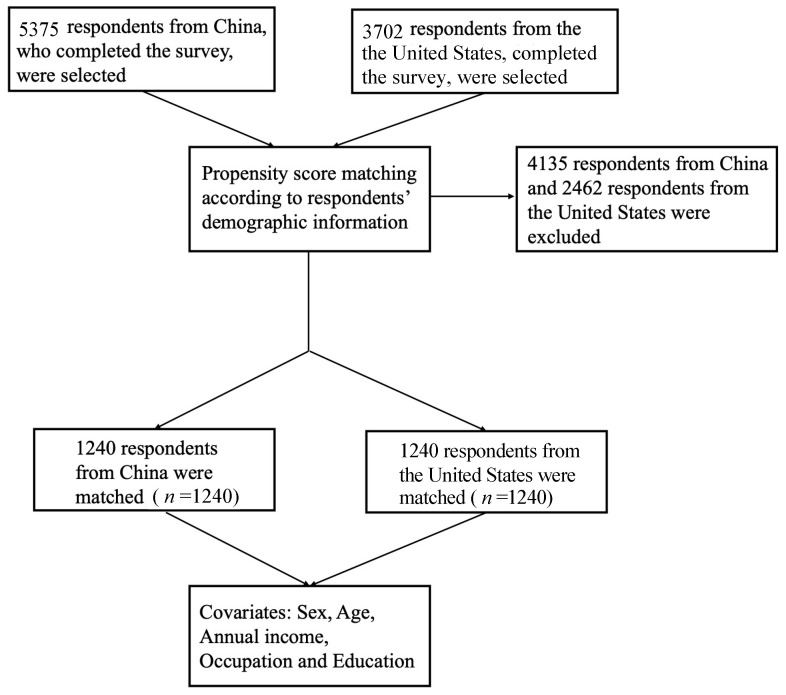
The flow chart of propensity score matching.

**Figure 2 vaccines-09-00649-f002:**
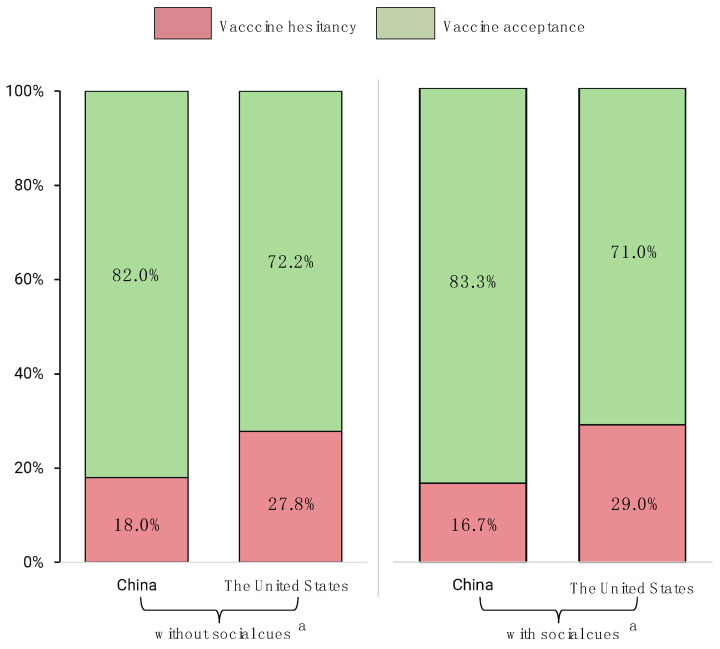
COVID-19 vaccination acceptance comparison between China and the United States after propensity score matching. ^a^ Social cues in this study means social factors that potentially impact respondents’ acceptance, i.e., recommendations from friends, family members, or employers, etc.

**Figure 3 vaccines-09-00649-f003:**
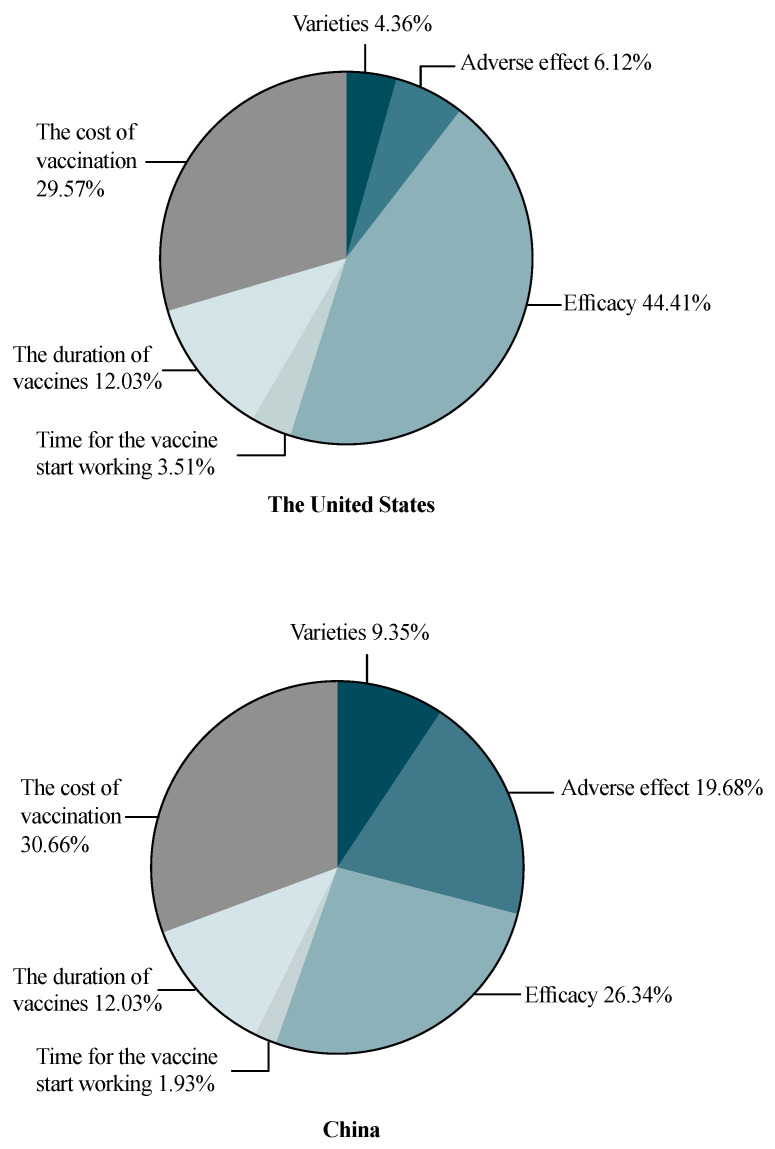
The relative importance of COVID-19 vaccine attributes: comparison between respondents from the United States and China.

**Table 1 vaccines-09-00649-t001:** Participants’ demographic information, a major source of information and acceptance.

Demographic Items	Unmatched			Matched		
	China	United States	*p*-Value	China	United States	*p*-Value
Sample size, n	5374	3701		1240	1240	
Sex (%)			0.003			1.00
Male	2385 (44.4%)	1765 (47.7%)		534 (43.1%)	535 (43.1%)	
Female	2971 (55.3%)	1918 (51.8%)		706 (56.9%)	705 (56.9%)	
Other	18 (0.3%)	18 (0.5%)		0 (0.0%)	0 (0.0%)	
Age interval in years (%)						
18–25	1119 (20.8%)	501 (13.5%)	<0.001	164 (13.2%)	162 (13.1%)	1.00
26–30	759 (14.1%)	762 (20.6%)		235 (19.0%)	234 (18.9%)	
31–35	697 (13.0%)	750 (20.3%)		244 (19.7%)	251 (20.2%)	
36–40	488 (9.1%)	505 (13.6%)		152 (12.3%)	152 (12.3%)	
41–45	514 (9.6%)	368 (9.9%)		139 (11.2%)	136 (11.0%)	
46–50	630 (11.7%)	241 (6.5%)		105 (8.5%)	103 (8.3%)	
51–55	429 (8.0%)	174 (4.7%)		89 (7.2%)	84 (6.8%)	
56–60	347 (6.5%)	154 (4.2%)		60 (4.8%)	60 (4.8%)	
Above 60	391 (7.3%)	246 (6.6%)		52 (4.2%)	58 (4.7%)	
Highest educational level (%)						
Pre-primary education or primary school education	399 (7.4%)	2 (0.1%)	<0.001	Non-postgraduate 1221 (98.5%)	Non- postgraduate 1222 (98.5%)	0.87
Middle school education	591 (11.0%)	15 (0.4%)				
High school education	933 (17.4%)	675 (18.2%)				
Vocational school education	887 (16.5%)	508 (13.7%)				
Bachelor’s degree	2021 (37.6%)	1710 (46.2%)				
Master’s degree	422 (7.9%)	711 (19.2%)				
PhD degree	121 (2.3%)	80 (2.2%)		19 (1.5%)	18 (1.5%)	
Occupation and working area (%)						
Students	1230 (22.9%)	249 (6.7%)	<0.001	139 (11.2%)	132 (10.6%)	1.00
Managers	681 (12.7%)	541 (14.6%)		178 (14.4%)	174 (14.0%)	
Professionals	768 (14.3%)	993 (26.8%)		250 (20.2%)	250 (20.2%)	
Technicians and associate professionals	795 (14.8%)	423 (11.4%)		148 (11.9%)	157 (12.7%)	
Clerical support workers	230 (4.3%)	318 (8.6%)		121 (9.8%)	122 (9.8%)	
Service and sales workers	515 (9.6%)	453 (12.2%)		185 (14.9%)	188 (15.2%)	
Skilled agricultural, forestry and fishery workers	373 (6.9%)	43 (1.2%)		14 (1.1%)	14 (1.1%)	
Craft and related trade workers	121 (2.3%)	78 (2.1%)		27 (2.2%)	28 (2.3%)	
Plant and machine operators and assemblers	184 (3.4%)	32 (0.9%)		11 (0.9%)	11 (0.9%)	
Elementary occupations	133 (2.5%)	75 (2.0%)		16 (1.3%)	14 (1.1%)	
Armed forces occupations	73 (1.4%)	19 (0.5%)		6 (0.5%)	4 (0.3%)	
Other	271 (5.0%)	477 (12.9%)		145 (11.7%)	146 (11.8%)	
Annual salary level (%)						
Under USD 10,000	2254 (48.0%)	398 (11.0%)	<0.001	335 (27.0%)	333 (26.9%)	1.00
USD 10,001–20000	1225 (26.1%)	382 (10.6%)		257 (20.7%)	259 (20.9%)	
USD 20,001–30,000	561 (12.0%)	481 (13.3%)		236 (19.0%)	236 (19.0%)	
USD 30,001–40,000	296 (6.3%)	472 (13.1%)		192 (15.5%)	193 (15.6%)	
USD 40,001–50,000	163 (3.5%)	456 (12.6%)		91 (7.3%)	92 (7.4%)	
USD 50,001–60,000	55 (1.2%)	464 (12.8%)		41 (3.3%)	40 (3.2%)	
USD 60,001–70,000	46 (1.0%)	331 (9.2%)		23 (1.9%)	24 (1.9%)	
Above USD 70,000	94 (2.0%)	630 (17.4%)		65 (5.2%)	63 (5.1%)	
Acceptance of vaccination (totally unwilling, 0–totally willing, 10)						
Average	7.8 (2.5)	7.2 (3.5)	<0.001	7.8 (2.4)	6.9 (3.5)	<0.001
Acceptance of vaccination if someone else recommends (totally unwilling, 0–totally willing, 10)						
Average	7.8 (2.5)	7.2 (3.3)	<0.001	7.8 (2.3)	6.9 (3.5)	<0.001
Ever been infected with COVID-19? (%)						
Yes	56 (1.0%)	569 (15.4%)	<0.001	21 (1.7%)	194 (15.6%)	<0.001
No	5243 (97.6%)	3102 (83.8%)		1196 (96.5%)	1033 (83.3%)	
Not answered	75 (1.4%)	30 (0.8%)		23 (1.9%)	13 (1.0%)	
Friend family or community ever infected? (%)						
Yes	149 (2.8%)	2575 (69.6%)	<0.001	47 (3.8%)	815 (65.7%)	<0.001
No	5110 (95.1%)	1096 (29.6%)		1156 (93.2%)	415 (33.5%)	
Not answered	115 (2.1%)	30 (0.8%)		37 (3.0%)	10 (0.8%)	
Marital state (%)						
Single	1830 (34.1%)	1391 (37.6%)	<0.001	372 (30.0%)	522 (42.1%)	<0.001
Married	3107 (57.8%)	1905 (51.5%)		739 (59.6%)	577 (46.5%)	
Divorced	185 (3.4%)	273 (7.4%)		58 (4.7%)	98 (7.9%)	
Other	67 (1.2%)	109 (2.9%)		16 (1.3%)	38 (3.1%)	
Not answered	185 (3.4%)	23 (0.6%)		55 (4.4%)	5 (0.4%)	
Source of information of COVID-19 vaccines (%)						
Healthcare provider	1844 (34.3%)	1294 (35.0%)	0.52	433 (34.9%)	387 (31.2%)	0.050
CDC or public health department	2464 (45.9%)	1958 (52.9%)	<0.001	583 (47.0%)	612 (49.4%)	0.24
News reports	3907 (72.7%)	2333 (63.0%)	<0.001	908 (73.2%)	773 (62.3%)	<0.001
Social media	2720 (50.6%)	1490 (40.3%)	<0.001	645 (52.0%)	505 (40.7%)	<0.001
Friends or family members	2065 (38.4%)	1199 (32.4%)	<0.001	431 (34.8%)	380 (30.6%)	0.029
Employers	422 (7.9%)	457 (12.3%)	<0.001	112 (9.0%)	110 (8.9%)	0.89
Pharmaceutical company advertisement	432 (8.0%)	123 (3.3%)	<0.001	100 (8.1%)	41 (3.3%)	<0.001
Other	14 (0.3%)	54 (1.5%)	<0.001	4 (0.3%)	17 (1.4%)	0.004

**Table 2 vaccines-09-00649-t002:** Share of preference and scenario analysis results.

Share of Preference	Base Scenario	Scenario 1	Scenario 2	Scenario 3	Scenario 4	Scenario 5	Scenario 6	Scenario 7	Scenario 8
*United States*									
Vaccine varieties	Inactivated vaccine	Adenovirus vaccine	mRNA	mRNA	Inactivated vaccine	Adenovirus vaccine	Adenovirus vaccine	Adenovirus vaccine	mRNA
Adverse effect	moderate	Very mild	Moderate	Mild	Very mild	Mild	Very mild	Very mild	Very mild
Efficacy	55%	75%	95%	95%	75%	95%	65%	75%	95%
Time for the vaccine to start working	20 days	20 days	20 days	20 days	10 days	20 days	5 days	5 days	5 days
The duration of vaccine effectiveness	5 months	5 months	5 months	5 months	5 months	5 months	5 months	5 months	20 months
The cost of vaccination	USD 50	USD 50	USD 50	USD 50	USD 50	USD 50	USD 50	USD 50	USD 50
Share of preference	2.9%	8.2%	14.2%	17.3%	7.7%	16.2%	5.2%	9.2%	19.1%
*China*									
Vaccine varieties	Adenovirus vaccine	Adenovirus vaccine	mRNA vaccine	mRNA vaccine	Inactivated vaccine	Adenovirus vaccine	Adenovirus vaccine	Adenovirus vaccine	Inactivated vaccine
Adverse effect	moderate	Very mild	Moderate	Mild	Very mild	Mild	Very mild	Very mild	Very mild
Efficacy	55%	75%	95%	95%	75%	95%	65%	75%	95%
Time for the vaccine to start working	20 days	20 days	20 days	20 days	10 days	20 days	5 days	5 days	5 days
The duration of vaccine effectiveness	5 months	5 months	5 months	5 months	5 months	5 months	5 months	5 months	20 months
The cost of vaccination	USD 50	USD 50	USD 50	USD 50	USD 50	USD 50	USD 50	USD 50	USD 50
Share of preference	4.5%	10.7%	9.4%	13.4%	13.7%	12.4%	7.9%	10.6%	17.6%

**Table 3 vaccines-09-00649-t003:** Comparison between the United States and China in attributes levels utility and odds ratios.

Attributes	Variable	Coefficient	Std. Error	OR	95% CI	*p*-Value	Coefficient	Std. Error	OR	95% CI	*p*-Value
		***United States (n = 1240)***					***China (n = 1240)***				
Varieties	mRNA	Reference									
	Adenovirus vector vaccines	0.006	0.019	0.940	(0.906–0.976)	0.761	−0.105	0.018	0.922	(0.890–0.954)	<0.001
	Inactivated vaccine	−0.073	0.019	0.869	(0.837–0.902)	<0.001	0.128	0.018	1.164	(1.124–1.205)	<0.001
Adverse effect	very mild	Reference									
	mild	0.070	0.019	1.013	(0.976–1.051)	<0.001	0.077	0.018	0.879	(0.849–0.910)	<0.001
	moderate	−0.128	0.019	0.831	(0.800–0.863)	<0.001	−0.284	0.018	0.612	(0.591–0.635)	<0.001
Efficacy	55%	Reference									
	65%	−0.516	0.028	1.213	(1.147–1.282)	<0.001	−0.246	0.025	1.088	(1.036–1.144)	<0.001
	75%	0.061	0.029	2.158	(2.038–2.285)	0.038	0.048	0.028	1.461	(1.384–1.543)	0.084
	85%	0.438	0.025	3.146	(2.994–3.306)	<0.001	0.204	0.024	1.707	(1.627–1.791)	<0.001
	95%	0.726	0.027	4.196	(3.980–4.424)	<0.001	0.326	0.026	1.928	(1.832–2.030)	<0.001
Time for the vaccine to start working	5 days	Reference									
	10 days	−0.035	0.024	0.904	(0.863–0.948)	0.142	0.021	0.022	1.025	(0.982–1.069)	0.324
	15 days	0.017	0.023	0.953	(0.912–0.996)	0.445	−0.027	0.022	0.976	(0.936–1.019)	0.217
	20 days	−0.048	0.025	0.893	(0.849–0.938)	0.060	0.008	0.024	1.011	(0.965–1.059)	0.732
The duration of vaccine effectiveness	5 months	Reference									
	10 months	−0.081	0.024	1.148	(1.096–1.202)	<0.001	−0.021	0.022	1.153	(1.105–1.204)	0.338
	15 months	0.131	0.024	1.419	(1.354–1.487)	<0.001	0.048	0.023	1.236	(1.182–1.292)	0.035
	20 months	0.169	0.023	1.475	(1.410–1.543)	<0.001	0.136	0.022	1.350	(1.293–1.409)	<0.001
The cost of vaccination	$0	Reference									
	$50	0.179	0.026	0.674	(0.640–0.709)	<0.001	0.162	0.025	0.776	(0.740–0.814)	<0.001
	$100	−0.111	0.026	0.504	(0.479–0.530)	<0.001	−0.029	0.024	0.641	(0.612–0.672)	0.225
	$150	−0.263	0.029	0.433	(0.409–0.458)	<0.001	−0.200	0.027	0.540	(0.513–0.570)	<0.001
	$200	−0.381	0.031	0.385	(0.362–0.409)	<0.001	−0.349	0.029	0.466	(0.440–0.493)	<0.001

## Data Availability

Data are available, upon reasonable request, by emailing: wkming@connect.hku.hk.

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
