# Peer review of "A Comparison of Vaccine Hesitancy of COVID-19 Vaccination in China and the United States"

_vaccines, 2021, doi:10.3390/vaccines9060649_

Round 1
Reviewer 1 Report
This is generally an interesting paper which, not surprisingly, indicates that community views towards vaccination differ in China and the US. The main take-home message is that vaccine rollout strategies should take heed of prevailing attitudes of the population in the affected countries.
To compare attitudes in China and the US has required statistical corrections using propensity score matching and although I'm not a statistics expert the methodology appears to follow generally accepted parameters.
Whilst I'm happy with the scientific quality of the paper the readability needs significant improvement in terms of grammar. Terms such as "a.k.a." and "etc" should not be used in scientific papers. Vaccination doses should not be referred to as "shots". I suggest that a native English-speaking editor reviews and corrects the numerous grammatical errors.
Although costs are highlighted as a concern for people both in the United States and China both governments have indicated that vaccines will be free of charge for their populations. Maybe this was not the case when the survey was being performed but would be worthwhile updating this information in the discussion.
Author Response
|Response to minor comment 1:
We thank the reviewer for the high compliment on our manuscript. We have revised our grammatical problems. And the following sentences are revised:
(Page 1 line 33-34) “Propensity score matching (PSM) was performed to enable a direct comparison between the two countries.”
(Page 1 line 34-37) “A total of 9,077 (5,375 and 3,702 from China and the US, respectively) respondents completed the survey. After propensity score matching, over 82.0% of respondents from China positively accept the COVID-19 vaccination, while 72.2% of respondents from the US positively accept it.”
(Page 1 line 45-47) “Also, respondents from China tend to be much more concerned about the adverse effect of vaccination (19.68% vs. 6.12%) and have a lower perceived severity of being infected with COVID-19.”
(Page 2 line 57-59) “A pneumonia-like disease outbreak, now named the 2019 Coronavirus disease (COVID-19) caused by a newly identified coronavirus, the severe acute respiratory syndrome (SARS)-CoV-2, has swept the globe in early 2020.”
(Page 7 line 290-298) “As the pre-PSM listed in Table 1 and Figure 2 shows, respondents from China had a relatively lower hesitancy of COVID-19 vaccines (7.8/10) than those from the US (7.2/10) when asked, “How do you rate your willingness and acceptance to get COVID-19 vaccination? (if the vaccines are generally available)” (general acceptance). For post-PSM, over 82.0% of respondents from China positively accept the COVID-19 vaccination, while only 72.2% of respondents from the US positively accept it, and the proportion change to 83.3% and 71.0%, respectively, if the respondents were recommended to get vaccinated by friends, family members or employers and so on.”
(Page 12 line 434-439) “Combined with their low perceived risks of susceptibility, this may hence lead to their placing greater importance on the attribute “cost” in the discrete choice experiment (Table 2 and Figure 2) and may also explain why the vaccination rate remains at a relatively low level in China, compared with some other countries in the world, such as the US, the UK and so on.”
(Page 14 line 504-506) “Additionally, China and the US have plenty of differences, especially in contextual influences, for instance, media environment, influential leaders, historical influences, policies, and cultural factors.”
|Response to minor comment 2:
We thank the reviewer has noticed the problem that COVID-19 vaccines are free of charge for both population from two countries, therefore we update and discuss more for this point in the discussion section as below:
(Page 12-13, line 440-444) “Although both populations from the US and China highlight the effect of the weight of cost in their preference, both governments have made relative policies to guarantee the COVID-19 vaccines were free of charge. Therefore, the effect of the cost of COVID-19 vaccines can be ignored to some extent when prompt vaccination.”
Reviewer 2 Report
Early in the pandemic, the authors of this work carried out an interesting study to probe issues surrounding vaccine hesitancy, amongst two countries: China and the USA. As much can be said to be different between these nations, the authors collected data from responses from over 9000 participants in total to evaluate similarities and differences in issues relating to vaccine hesitancy.
The study design appears to be well organized with appropriate methods employed. The Tables are useful to describe the groups evaluated. The survey data was collected over a period of early February, when the pandemic was being felt in full burden across the globe, and vaccines were recently being rolled out. The study was pertinent to the worldwide populations of potential vaccination, as most to the world did not yet have access to any vaccination options.
While the questionnaire was discussed in detail, how it was distributed to populations in both China and the USA was not clearly delineated. As a result, it is hard to know how impartial the distribution was in both countries. Also, no racial classifications were described, which is important for distribution.
The statistical analysis appears to be appropriate and well performed.
A concern I have is that the authors conclusions about hesitancy have not been placed in the context the current status of vaccination in either country. The numbers of partially and fully vaccinated individual currently in much higher that when this report was initially submitted, so hesitancy within the population is likely very different and due to entirely different motivations than initially found. Please touch upon this to make this report relevant in the current context.
Concerns:
Sourcing and details of study design and respondents in US is not described. Must be clarified.
Some text issues:
79-81 Clarify
92-99 Run on sentence
134 COVID-10?
474-476 Unclear statement
481 Unclear statement
Author Response
| Response to minor comment 1: While the questionnaire was discussed in detail, how it was distributed to populations in both China and the USA was not clearly delineated. As a result, it is hard to know how impartial the distribution was in both countries. Also, no racial classifications were described, which is important for distribution.
We thank the reviewer for the high compliment for our manuscript. We have further elaborated the method of survey distribution in the following sentences.
Secondly, indeed, we have not collected information regarding race and ethnicity during the survey, hence the racial classifications were not available in our study, and we have addressed this in the limitation as follows:
(Page 3 line 125) “Stratified sampling by age and geological locations was used, and nationally representative samples of the general adult populations were collected in China, while the US respondents were mainly collected via MTurk32”
(Page 15, line 524-534) “Moreover, in view of impartiality, we did not collect information regarding race and ethnicity in the survey. Hence the racial classification of the respondents was not reported and considered in the analysis in the present study. Regarding survey distribution, the present study utilized multiple means to collect data, where the Chinese respondents were sampled using stratification sampling via age, and geographical location, while the US respondents were sampled mainly through MTurk. Despite that we have evaluated the reliability of the data and sampling adequacy using Kaiser, Meyer and Olkin analysis (KMO coef=0.8958), and that several studies have reported the census-level data retrieved from MTurk, the results should still be interpreted with caution.”
| Response to minor comment 2: A concern I have is that the authors conclusions about hesitancy have not been placed in the context the current status of vaccination in either country. The numbers of partially and fully vaccinated individual currently in much higher that when this report was initially submitted, so hesitancy within the population is likely very different and due to entirely different motivations than initially found. Please touch upon this to make this report relevant in the current context. Sourcing and details of study design and respondents in US is not described. Must be clarified.
We agreed with the reviewer. Despite that the acceptance of the COVID-19 vaccination may be ever-changing, the present study has investigated the factors underlying the acceptance and preference, which may provide some insights in future vaccine promotion. And we have further clarified the concern in the limitation as follows:
(Page 15 line 518) “The present study has limitations. The inherent nature of cross-sectional studies renders it difficult to establish causality or to generalize the results in a long-term manner, especially when vaccination acceptance is variable, dynamic and multifactorial. Hence the results of the study should be interpreted with caution. But the present study implemented the PSM to minimize the confounding effects, and it may somehow enhance the interpretability of the results.”
| Response to minor comment 3:
Some text issues:
79-81 Clarify
92-99 Run on sentence
134 COVID-10?
474-476 Unclear statement
481 Unclear statement
Thanks. We have revised the corresponding text as follows:
(Page 3 line 79-81) “Therefore, massive vaccination coverage is considered as a prerequisite to achieve herd immunity and therefore curb the COVID-19 pandemic14. By March 2021, several COVID-19 vaccines15-17 have been developed, and vaccination rollout has started in a few countries.”
(page 3 line 94) “Some studies have found that COVID-19 vaccine acceptance varied to a large extent across countries and regions before the vaccine becomes available; the vaccine hesitancy of COVID-19 is increasing globally25,26. ”
(Page 14 line 480-482) “Formal collaboration has been institutionalized through the World Health Organization (WHO); however, in this COVID-19 pandemic, many countries showed poor performance against COVID-19 with the refection of strong self-interested nationalism55.”
(Page 14 line 485-487) “Meanwhile, vaccine nationalism has been warned by WHO as a moral failure. Therefore, to relieve the global epidemic, collaboration on COVID-19 vaccines have been urgently needed, especially when some adenovirus COVID-19 vaccination has been suspended in many countries due to adverse effects56.”